# Self-Perception and Self-Acceptance Are Related to Unhealthy Weight Control Behaviors in Catalan Adolescents: A Cross-Sectional Study

**DOI:** 10.3390/ijerph18094976

**Published:** 2021-05-07

**Authors:** Mercè Pollina-Pocallet, Eva Artigues-Barberà, Glòria Tort-Nasarre, Joaquim Sol, Laura Azlor, Quintí Foguet-Boreu, Marta Ortega-Bravo

**Affiliations:** 1Bellpuig Primary Care Center, Gerència Territorial Lleida, Catalan Health Institute (ICS), Diputació, 5, 25250 Bellpuig, Lleida, Spain; mpollina.lleida.ics@gencat.cat; 2Department of Nursing, Faculty of Nursing and Physiotherapy, University of Lleida, Carrer de Montserrat Roig, 25198 Lleida, Spain; eartigues.lleida.ics@gencat.cat (E.A.-B.); gtort.cc.ics@gencat.cat (G.T.-N.); 3Catalan Health Institute (ICS), Primary Care Lleida, Rambla Ferran, 44, 25007 Lleida, Spain; jsol.lleida.ics@gencat.cat; 4Research Support Unit Lleida, Fundació Institut Universitari per a la Recerca a l’Atenció Primària de Salut Jordi Gol i Gurina (IDIAPJGol), Rambla Ferran, 44, 25007 Lleida, Spain; lazlor.lleida.ics@gencat.cat; 5Research Group in Therapies in Primary Care (GRETAPS), Rambla Ferran, 44, 25007 Lleida, Spain; 6Health Education Research Group (GREpS), Faculty of Nursing and Physiotherapy, University of Lleida, Carrer de Montserrat Roig, 25198 Lleida, Spain; 7Calaf Primary Care Center, Gerència Territorial Catalunya Central, Catalan Health Institute (ICS), Cta. Llarga 19, 08280 Calaf, Barcelona, Spain; 8Metabolic Physiopathology Group, Department of Experimental Medicine, University of Lleida-IRBLleida, Avinguda Alcalde Rovira Roure, 80, 25198 Lleida, Spain; 9Central Research Unit, Fundació Institut Universitari per a la Recerca a l’Atenció Primària de Salut Jordi Gol i Gurina (IDIAPJGol), Gran Via Corts Catalanes, 587, àtic, 08007 Barcelona, Spain; 42292qfb@comb.cat; 10Department of Psychiatry, Vic University Hospital, Francesc Pla el Vigatà, 1, 08500 Vic (Barcelona), Spain; 11Faculty of Medicine, University of Vic-Central University of Catalonia (UVic-UCC), 08500 Vic (Barcelona), Catalonia, Spain; 12Faculty of Medicine, University of Lleida, Carrer de Montserrat Roig, 225198 Lleida, Spain

**Keywords:** adolescence, body image, body dissatisfaction, adolescent behavior

## Abstract

Adolescence is associated with a higher vulnerability that may result in a high dissatisfaction, the practice of unhealthy weight-control behaviors (UWCB) and, eventually, the onset of body image-related mental disorders. These factors are strongly associated with the social context, so it is important to characterize them in local or regional studies. To assess the relationship between body image and UWCB presence, a cross-sectional study was performed among 2496 schooled adolescents from Lleida (Spain) between 2017 and 2019. Their perceived and desired images were evaluated and compared with the real image in order to obtain the body distortion and the body dissatisfaction and relate them with UWCB. The studied individuals perceived themselves thinner than they actually were, with no differences between males and females. However, differences were found regarding body dissatisfaction, showing that females desired to be thinner, while males desired a more corpulent body image. Furthermore, one out of ten individuals reported UWCB, with higher prevalence among females. UWCB was associated with a desire to be thinner and with distorted body images. It is essential to work on self-perception and self-acceptance in early adolescence from an interdisciplinary perspective at educational, social and health levels to promote health in adolescence.

## 1. Introduction

Adolescence is a transition period between childhood and adult age and can be divided into early (from 10 to 14 years old) and late adolescence (from 15 to 19) [1]. This period is characterized by important physical, psychological and social changes [2,3]. Several social agents have a strong interaction with the adolescent, such as family, peer groups, educational institutions and media and build a system of values and beliefs that are the basis for the adolescent self-confidence [4]. The quality of these relationships has a key role in the psychosocial development [5]. It does not affect neither male nor female in the same way [6] and could lead to adulthood mental disorders, getting to the estimate that half of these mental disorders start at adolescence [1]. 

One of the most commonly developed mental disorders at this age is the body dysmorphic disorder (BDD) [7], which is a severe dysfunction related to the obsessive compulsive disorder [8] and characterized by an impaired preoccupation about the appearance [9] that is associated with compulsive behaviors, impairment in social functioning, psychiatric hospitalizations and suicide attempts [10,11].

BDD is strongly related to body image [12], which was firstly described only as a mental representation of the own body [13], but later was considered as a multidimensional construct [14]. It can be divided into three main dimensions that are hierarchically related: the perceptual dimension, related to the precision in body volume perception, the cognitive-affective dimension, referring to those thoughts and feelings related to this perception and the behavioral dimension, which comprises the actions and behaviors derived from the subjective feelings [15]. Affections in the perceptual dimension are related to a distortion by either an overestimation or an underestimation of the body size; affections in the cognitive-affective dimension are related to body dissatisfaction and these could lead to affections in the behavioral dimension, which are associated with unhealthy weight-control behaviors (UWCB), such as unsupervised diets, self-induced vomiting, laxative, diuretic or weight loss products intake and fasting [16], either with or without help of some other factors, such as exercise [17]. 

Due to sociocultural reasons, preadolescents and adolescents lean toward a certain kind of aesthetic, causing them to show excessive concern for their body, with an ‘ideal figure’, which was originated in Western societies and has a high influence in Eastern ones [18], where thinness is enhanced. This fact has led them, especially adolescent and young adult females, to have disordered eating habits [19,20]. Perception of beauty, critical appraisal and media literacy, along with other internal factors, such as people’s personality and ways of being [21], attitude toward self and toward others [22], as well as one’s own cognitive strategies [23], have been associated with self-perception of body image. An inaccurate self-perception of body weight can also increase the risk of weight preoccupations and weight control among normal-weight adolescents [24]. Dissatisfaction with body image during the early stages of adolescence has also been linked to a poorer self-esteem and might advance depressive symptoms, as well as a higher body mass index, a reduction of physical activity, a poorer dietary quality, disordered eating and clinical eating disorders [25]. Previous studies have identified that weight control behaviors may differ according to either socioeconomic [15], or sex [26] and weight status, particularly among those who are overweight or obese [27] and the onset of this behavior at younger ages may even increase that risk [17]. In this sense, rural populations are more likely to be obese and overweight compared to their urban counterparts; probably due to the limited availability of healthy food in households, as well as individual characteristics and preferences [28] and are usually underrepresented in studies [29]. In a 2011 study, Vander observed that low life satisfaction, high negative affectivity and body size dissatisfaction were associated with unhealthy weight control behaviors among boys, just as low life satisfaction was for girls [30]. UWCBs can be related to a risk of developing an eating disorder [31], being the most common between adolescent population nervous anorexia, bulimia and unspecified eating disorders (binge eating) [32].

Due to the importance of addressing these and other health issues at an early age, the Catalan Government initiated the “Health and School Program” in 2004. This program was a collaborative project between the Health Department and the Education Department with the aim to improve the health of the adolescents through the inclusion of health professionals in the school environment and their active implication in health promotion and prevention [33].

Taking into account these premises and the importance of the social environment in body image, we thought that it would be mandatory to obtain data from adolescents who reflected a mostly rural and disperse community, such as Lleida (Spain). The current study was designed to evaluate the self-perception, body distortion and dissatisfaction among adolescents and assess their relationship with UWCB presence, including self-induced vomiting, laxative, diuretic and/or weight loss products intake and fasting. 

## 2. Materials and Methods

### 2.1. Study Design and Participants

Cross-sectional study among adolescents (from 12 to 19 years old) from High Schools and vocational schools of Lleida (Spain) between 2017 and 2019. All students in the age range who willingly agreed to participate were included, except those with an intellectual disability or linguistic barriers which prevented them from fulfilling the questionnaire, adding up to a total of 2496 participants. Those schools included in the “Health and School Program” incorporated the study as a part of the program. 

### 2.2. Data Collection and Tests

Data collection was performed in collaboration with the PICA-AD work group (Appendix A). The project was presented to the school boards and fieldwork dates were established with those that wanted to participate. Informed consents were given to the students and they were asked to return them the day before the data collection. Data regarding age, sex and anthropometric measures were collected by trained healthcare professionals. Height and weight were obtained using a Roman scale and a stadiometer, respectively, from participants wearing a layer of underwear and not wearing shoes. The same day, data regarding the three dimensions of the body image were obtained from a self-reported questionnaire. The questionnaires were responded inside the classroom, before the beginning of the class and were supervised by the classroom teacher and the responsible of the data collection. All the students from the same class did them at the same time.

#### 2.2.1. Perceptual and Cognitive-Affective Dimensions

The perceptual dimension (precision in body volume perception) was evaluated in terms of real image, perceived image and body distortion. The cognitive affective dimension (feelings related to the perception) was evaluated in terms of desired image and body dissatisfaction. For this purpose, a validated 13-card scale (13-CS) was used [34]. The scale consists in 13 schematic images representing human figures. The middle image represents the reference population median BMI and the other ones are obtained by stepwise increasing or reducing the reference volume up to ±30%, in intervals of 5%. Figure values ranged from −6 to 6, with the rate 0 assigned to the middle image.

Participants were asked to choose the image they believed best represented them (perceived image) and the one which they would like the more to seem (desired image).

A 13-CS “real image” was assigned to participants according to their age and sex standardized body mass index (BMI) percentiles, which were calculated using the World Health Organization’s growth reference tables [35]. Since the 13-CS images are proportionally distributed, a proportional percentile (in intervals of 7.7%) was assigned to each image. 

Body distortion score (distortion) was calculated as the difference of images between the 13-CS perceived image and the real image. Body dissatisfaction score (dissatisfaction) was established as the difference between the perceived image and the desired image. Scores ranged from −12 to 12. Negative scores meant that the perceived image was lower than the real/desired one, while positive scores indicated that the perceived image was higher than the real/desired one.

#### 2.2.2. Behavioral Dimension

In order to evaluate the behavioral dimension (behaviors derived from the feelings towards the own body), participants were asked about UWCB (self-induced vomiting, laxative or diuretic products and/or weight loss pills intake and fasting) using an adaptation of the Eating and Activity in Teens (EAT) Survey [36], which consisted in five direct questions with dichotomous response (yes/no) about whether the participants had had any of the mentioned behaviors in the last 12 months. 

### 2.3. Statistical Analysis

A statistical description of the study variables was performed, taking into consideration all data, sex and age subsets. Participants were divided into two age groups (early adolescents, <15 years old and late adolescents, ≥15 years old) according to the UNICEF classification [1]. This is also the age in which adolescents are no longer attended in pediatric medicine, according to the Primary Care organizational system of the Autonomous Region of Catalonia (Spain). Numerical variables were expressed as mean and standard deviation and categorical variables were expressed as absolute and relative frequencies. One sample Student’s T-tests or Mann–Whitney U tests (depending on whether variables were normally distributed or not, respectively, according Shapiro–Wilks test) were performed to all the data and its sex and age subsets regarding their real, perceived and desired images, as well as to dissatisfaction and distortion, in order to determine whether they were significantly different than 0. 

For testing differences between sex and age groups regarding the perceptual and cognitive-affective dimensions, Student’s T-tests or Mann–Whitney U tests were applied depending on normality of the variables. For testing differences between sex and age groups regarding the behavioral dimension, chi squared tests were applied for the variables “presence of one or more UWCB”, “self-induced vomiting”, “laxatives intake”, “diuretics intake”, “weight loss pills intake” and “fasting”. The same variables were evaluated using random-intercept mixed effects logistic regressions in order to test whether they were associated either to distortion and/or dissatisfaction. The “school” variable was considered as the cluster variable and body dissatisfaction and body distortion index were used as polynomial predictor variables, altogether with age and sex. The models were adjusted by BMI and interactions between the scores and sex were also considered. Statistical significance was set at *p* < 0.05 without adjustment for multiple comparisons [37]. The analysis was performed using R statistical software, version 3.6.1.

## 3. Results

A total of 45 out of 66 schools agreed to participate. From the 12,754 students from these schools, 2500 (19.6%) gave their consent to participate. 4 participants were over 19 years old and were, therefore, excluded. Hence, the analysis was performed on the 2496 remaining participants. 51.2% of them were females and 63.8% were early adolescents (Table 1). 

### 3.1. Perceptual Dimension 

The mean real image of the studied population according to the 13-CS images was slightly higher than 0. Taking into account that figures range from −6 to 6 and that the median of the reference population is assigned to figure 0, the mean (SD) value of the population of our study was 1.05 (3.64). Females were thinner than males and late adolescents were also thinner than early adolescents (Table 2, Figure 1). 

Although the mean real images were higher than 0 in all the data subsets, the means of the reported perceived images were always lower than 0, indicating that students perceived themselves thinner than the median of the reference population. Females perceived themselves thinner than males (Table 2, Figure 1). 

When both images were related using the body distortion score, we observed that the participants reported lower perceived images with respect to their real image. In terms of distortion index, the mean (SD) score was −3.47 (2.92). We did not find significant differences between males and females. However, late adolescents had significantly higher distortion indexes, which, in this case, corresponded to more realistic images (Table 2). 

### 3.2. Cognitive-Affective Dimension

In order to explore the cognitive-affective dimension, students reported their desired image and body dissatisfaction scores were calculated. The mean (SD) desired image was −2.79 (2.01), indicating that the studied population desired images are lower than the median BMI of the reference population. This desired image was lower in females (Table 2, Figure 1). 

The studied population reported lower desired figures than the perceived ones, resulting in a mean (SD) dissatisfaction score of 0.37 (2.22). Despite the global positive dissatisfaction scores, males obtained significantly lower scores than females, which resulted in negative dissatisfaction scores in males with a mean (SD) of −0.25 (2.35) and positive dissatisfaction scores in females, with a mean (SD) of 0.96 (1.90). This fact indicated that males desired images with higher volumes than the perceived ones, while females desired thinner images. Early adolescents had higher dissatisfaction scores than late adolescents (Table 2).

### 3.3. Behavioral Dimension

Among all the participants, 10.4% had had one or more UWCB in the last 12 months. 2.61% from the total reported self-induced vomiting, 0.92% reported laxatives intake, 0.44% diuretics intake, 0.44% weight loss pills intake and 8.26% reported fasting. A significantly higher number of females than males reported UWCB and, specifically, this prevalence was higher in self-induced vomiting and fasting. No differences were found between age groups (Table 2).

When analyzing the associations between age, sex and the calculated scores and the presence of UWCB (adjusting by BMI), we found a significant linear association between the presence of UWCB and dissatisfaction score and a quadratic association between the presence of UWCB and the distortion score, indicating that a higher probability of having these behaviors was observed in individuals with higher dissatisfaction scores (desire to be thinner) and with distortion scores further from 0, either positive or negative. Males with high dissatisfaction scores had lower probability of UWCB than females; however, the probability for a score of 0 was the same. No associations were found with age (Table 3, Figure 2).

The associations were also evaluated for the specific UWCB. Similar results were found for self-induced vomiting and fasting. However, there was no association between fasting and the interaction between sex and body dissatisfaction and a positive association was found between fasting and age (Table 3, Figure 3 and Figure 4). No associations were found for laxatives, diuretics or weight-loss pills intake.

## 4. Discussion

Eating disorders are a recurrent issue between adolescents and are related to affections in the body perception and the presence of unhealthy weight control behaviors. In order to address this issue, different studies have been performed in Spain [38,39,40,41,42], but they are mainly located in urban areas, focused on dieting and physical exercise and use diverse methodologies. Therefore, different results are obtained. To the best of our knowledge, this is the first study of these dimensions performed in a Spanish rural area, which may have higher risk of UWCB due to the expected higher prevalence of overweight and obesity [28].

The present study describes a high prevalence of UWCB between adolescents. It also demonstrates that these adolescents tend to underestimate their body size and reports they have a dissatisfaction with the own body. Furthermore, it describes an association between UWCB with the desire of achieving thinner images and with distorted body perceptions, regardless the type of distortion.

A significant amount of the adolescents in the studied region have a distorted perception of their own body. The mean distortion obtained was around −3, which indicates that the perceived image is 3 figures thinner than the real. Other studies, including a pilot study performed in the region by our research group [43], have also reported an underestimation of the body image [43,44,45], which could be caused by a tendency to consider oneself in unrealistically positive terms [44] or by the increased prevalence of overweight and obesity, which could lead to an association of this type of figures with normality [46]. However, other studies have obtained opposite [24,47,48,49,50] or neutral results [51]. 

Regarding dissatisfaction, results from our previous pilot study are validated in the current study (females want to have thinner images and males desire more body volume) [43]. Although a desire to be thinner has been historically associated to females, our study reinforces the idea that dissatisfaction is expressed in a different way in males and might be associated with a desire of stronger bodies [52], as other authors report in other environments [40,47]. These patterns correspond to the beauty ideals transmitted by mass media and social networks [53]. In this line, other authors have associated dissatisfaction to the exposure to social networks, magazines or other media related to body image [54,55,56,57]. It is noteworthy that although females already underestimate their body size (they perceive themselves thinner than they actually are), they desire even thinner images. Regarding age, the late adolescent group has less dissatisfaction than the early adolescent one, which may be related to the own maturing process, which would help them to acquire confrontation strategies to neutralize the external pressure and increase their satisfaction in different aspects, such as body image [58].

If we focus on UWCB, one out of twelve teenage boys and one out of eight girls have had at least one of the studied behaviors (vomiting, laxatives, diuretics, pills, or fasting) in the past 12 months. If we contextualize this data in a Spanish class of 30 students [59], three of them (approximately two females and one male) would have an UWCB. Prolonged fasting was the most common UWCB among the studied adolescents, as previously described [30,60], followed by self-induced vomiting. The high prevalence of this type of behaviors is worrisome, as it is directly related to more serious problems such as anorexia, bulimia or other eating and mental disorders [8,61], which are, at the same time, associated with a high mortality and, in females, with future growth problems, anxiety, depression, pregnancy complications and unhealthy behaviors such as smoking or drug use [62,63].

It is noteworthy that males have also a high prevalence of UWCB and only differ from females at high dissatisfaction values. However, many studies have focused on eating disorders in females and it is important to remark that males also have this risk and may suffer from additional stigma due to the association of this disorder with the female sex [64].

An association has been found between dissatisfaction and distortion with UWCB. The likelihood of UWCB increases with high dissatisfaction (desired silhouettes thinner than the perceived ones) and with high distortions (overestimation or underestimation of body image). 

Body dissatisfaction and distortions have been previously related to unhealthy habits, especially in adolescents and in several cases these behaviors remain until adulthood [65]. The current “beauty standards” have a high impact into this population because they are continuously being exposed to them in television and social media [66]. This exposure, combined to the vulnerability derived from the physical and mental changes occurring during adolescence, may lead to a need to fit into these standards at any cost [66]. In this sense, concerns about weight, body satisfaction and depressive symptoms have been described as predictors of eating disorders in a person after 10 years [27]. Other studies have also described a short and long-term recurrence of behaviors initiated in adolescence, as well as an increased risk of developing anxiety and depression in adolescents with subclinical conditions [65,67,68,69]. Thus, the fact that acquired behaviors persist into adulthood is of vital importance, as it reinforces the idea that acquired education and habits are crucial to achieving healthy maturation and aging, both physically and mentally. Therefore, this is a matter of concern and, from our point of view, a matter of public health, given the results of our research in all these areas. On the other hand, the fact that an adequate environment (in both family and school) plays a protective role and promote self-esteem, security and confidence [70], is hopeful and makes it even more important to conduct regional studies like the current one and their subsequent interventions in these environments. In this sense, the “Health and School Program” provides the ideal environment for the implementation of a future intervention regarding self-perception and self-acceptance in order to address these issues. 

The study has some limitations. In the first place, dieting was not included as an UWCB because of the difficulty in discerning whether the diet was healthy or not, so we decided to focus on more severe UWCB. Secondly, the dissatisfaction scores were assumed rather than directly asked and the more corpulent images of the scales used might not be associated to fat images. Thirdly, age might be not representative of mental maturity and other measures might have been appropriate in order to complement the results. Finally, participation was voluntary and UWCB were self-declared, which could lead to a selection bias and the results could underestimate the real prevalence. However, this reinforces the severity and magnitude of the obtained results. Furthermore, the study included a large sample that was representative of a whole territory and obtained similar results as the pilot study, indicating the robustness of the study.

## 5. Conclusions

The study shows that adolescents are perceived to be thinner than they actually are and that, despite this, females want to be thinner and males more corpulent. The risk of having UWCB has been associated with body dissatisfaction and distortion. It is essential to work on self-perception and self-acceptance in early adolescence from an interdisciplinary perspective at educational, social and health levels to promote health in adolescence and indirectly in adulthood.

## Figures and Tables

**Figure 1 ijerph-18-04976-f001:**
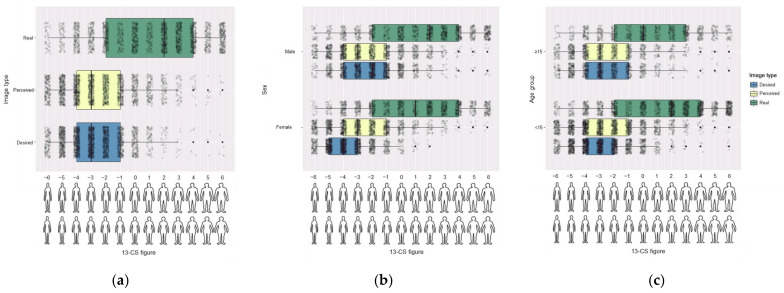
Boxplots of the real, perceived and desired body images according to a 13-Card Scale (13-CS) (**a**) including all the participants; (**b**) stratifying by sex; (**c**) stratifying by age group.

**Figure 2 ijerph-18-04976-f002:**
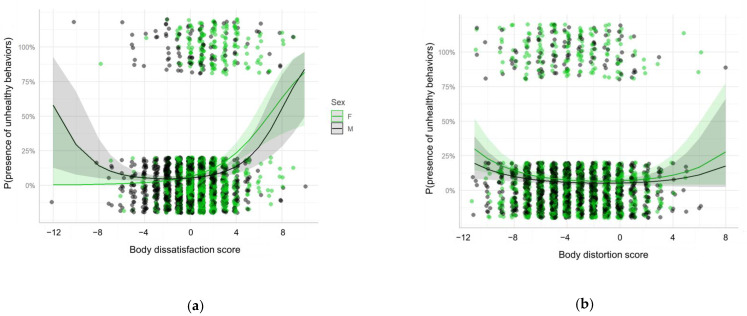
Random-intercept mixed effects logistic regression model for the presence of unhealthy weight control behaviors as a function of (**a**) Body dissatisfaction score; (**b**) Body distortion score; and their interaction with sex. Each point represents a participant, the lines represent the fitted model and the shades represent the 95% confidence interval. The points are slightly jittered in order to avoid overplotting. F: Female; M: Male.

**Figure 3 ijerph-18-04976-f003:**
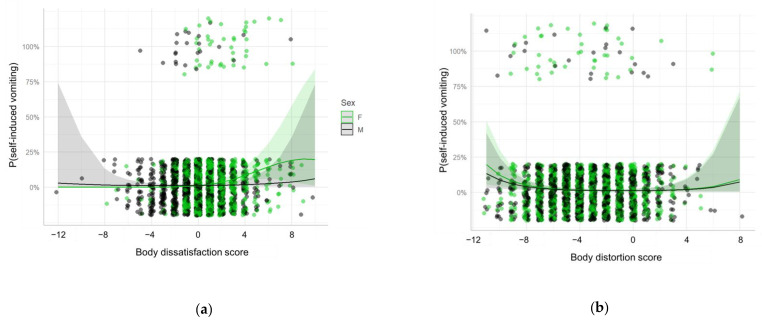
Random-intercept mixed effects logistic regression model for the presence of self-induced vomiting as a function of (**a**) Body dissatisfaction score; (**b**) Body distortion score; and their interaction with sex. Each point represents a participant, the lines represent the fitted model and the shades represent the 95% confidence interval. The points are slightly jittered in order to avoid overplotting. F: Female; M: Male.

**Figure 4 ijerph-18-04976-f004:**
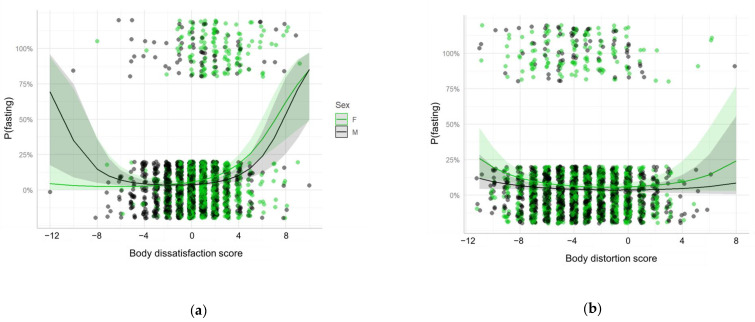
Random-intercept mixed effects logistic regression model for the presence of fasting as a function of (**a**) Body dissatisfaction score; (**b**) Body distortion score; and their interaction with sex. Each point represents a participant, the lines represent the fitted model and the shades represent the 95% confidence interval. The points are slightly jittered in order to avoid overplotting. F: Female; M: Male.

**Table 1 ijerph-18-04976-t001:** Descriptive table of the participants.

Item	All Participants (*n* = 2496)
Age, mean (SD)	14.1 (1.61)
Age group, *n* (%)	
<15	1593 (63.8%)
≥15	903 (36.2%)
Sex, *n* (%)	
Female	1278 (51.2%)
Male	1218 (48.8%)
Body Mass Index z-score, mean (SD)	0.31 (1.02)

**Table 2 ijerph-18-04976-t002:** Evaluation of the perceptual, cognitive affective and behavioral dimensions of the participants (*n* = 2496).

Item	All Participants (*n* = 2496)	Female (*n* = 1278)	Male (*n* = 1218)	*p*-Value (Male vs Female)	<15 (*n* = 1593)	≥15 (*n* = 903)	*p*-Value (<15 vs ≥15)
	Perceptual dimension						
Real image (*n* = 2496)	1.05 (3.64) *	0.82 (3.58) *	1.30 (3.68) *	<0.001	1.35 (3.71) *	0.52 (3.45) *	<0.001
Perceived image (*n* = 2474)	−2.42 (2.42) *	−2.61 (2.37) *	−2.22 (2.47) *	<0.001	−2.42 (2.42) *	−2.42 (2.43) *	0.834
Body distortion score (*n* = 2474)	−3.47 (2.92) *	−3.41 (2.88) *	−3.53 (2.97) *	0.161	−3.77 (3.00) *	−2.94 (2.71) *	<0.001
	Cognitive-affective dimension						
Desired image (*n* = 2473)	−2.79 (2.01) *	−3.57 (1.60) *	−1.98 (2.07) *	<0.001	−2.86 (1.95) *	−2.67 (2.10) *	0.053
Body dissatisfaction score (*n* = 2473)	0.37 (2.22) *	0.96 (1.90) *	−0.25 (2.35) *	<0.001	0.44 (2.26) *	0.25 (2.13) *	0.021
	Behavioral dimension						
Presence of unhealthy behavior (*n* = 2484)	257 (10.3%)	160 (12.6%)	97 (7.98%)	<0.001	161 (10.1%)	96 (10.7%)	0.712
Self-induced vomiting (*n* = 2489)	65 (2.61%)	43 (3.38%)	22 (1.81%)	0.020	43 (2.70%)	22 (2.45%)	0.803
Laxatives intake (*n* = 2490)	23 (0.92%)	11 (0.86%)	12 (0.99%)	0.914	17 (1.07%)	6 (0.67%)	0.429
Diuretics intake (*n* = 2488)	11 (0.44%)	6 (0.47%)	5 (0.41%)	1.000	10 (0.63%)	1 (0.11%)	0.111
Weight loss pills intake (*n* = 2492)	11 (0.44%)	7 (0.55%)	4 (0.33%)	0.598	8 (0.50%)	3 (0.33%)	0.756
Fasting (*n* = 2491)	205 (8.23%)	135 (10.6%)	70 (5.75%)	<0.001	122 (7.67%)	83 (9.22%)	0.201

* One-Sample Mann-Whitney U Test *p*-value < 0.05, indicating that the mean of the group is significantly different from 0.

**Table 3 ijerph-18-04976-t003:** Association between age, sex, body dissatisfaction and body distortion with unhealthy weight control behaviors (*n* = 2496).

Variable	Presence of UWCB		Self-Induced Vomiting		Fasting	
Predictor	aOR (95% CI)	*p*-value	aOR (95% CI)	*p*-value	aOR (95% CI)	*p*-value
Age	1.070 (0.979–1.170)	0.136	0.977 (0.825–1.148)	0.780	1.154 (1.048–1.269)	0.003
Sex (Male)	0.808 (0.476–1.371)	0.430	1.233 (0.453–3.389)	0.681	0.632 (0.338–1.157)	0.142
Body dissatisfaction (linear component)	1.406 (1.174–1.682)	<0.001	1.965 (1.352–3.021)	0.001	1.291 (1.101–1.565)	0.005
Body dissatisfaction (quadratic component)	1.008 (0.979–1.037)	0.605	0.964 (0.905–1.015)	0.209	1.020 (0.991–1.046)	0.155
Body distortion (linear component)	1.056 (0.919–1.212)	0.443	1.041 (0.837–1.276)	0.709	1.056 (0.907–1.224)	0.471
Body distortion (quadratic component)	1.019 (1.004–1.035)	0.016	1.029 (1.006–1.052)	0.009	1.019 (1.002–1.036)	0.024
Sex (Male): Body dissatisfaction (linear component)	0.804 (0.669–0.967)	0.021	0.538 (0.345–0.797)	0.004	0.868 (0.713–1.029)	0.132
Sex (Male): Body dissatisfaction (quadratic component)	1.025 (0.994–1.058)	0.119	1.048 (0.984–1.122)	0.154	1.018 (0.989–1.052)	0.249
Sex (Male): Body distortion (linear component)	0.991 (0.829–1.184)	0.918	1.000 (0.772–1.278)	0.998	0.968 (0.790–1.182)	0.750
Sex (Male): Body distortion (quadratic component)	0.997 (0.975–1.019)	0.781	0.996 (0.965–1.028)	0.815	0.993 (0.969–1.018)	0.592

UWCB: Unhealthy weight control behaviors; aOR: Adjusted Odds Ratio; CI: confidence interval.

## Data Availability

The data is hosted on the research team internal servers and will beprovided under reasonable request.

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
