# Peer review of "Self-Perception and Self-Acceptance Are Related to Unhealthy Weight Control Behaviors in Catalan Adolescents: A Cross-Sectional Study"

_ijerph, 2021, doi:10.3390/ijerph18094976_

Round 1
Reviewer 1 Report
Brief remarks:
This work reviews the relationship between expected and desired body types in young people in relation to unhealthy weight control behaviours. The researchers state that there is a relationship between perceiving oneself as thinner than is true and also desiring a thinner (in females) or larger (in males) body type and unhealthy weight control behaviours. The research is interesting and considered, and builds on previous work by this group. Nevertheless, this reviewer raises a few queries and improvements that could be made in order to strengthen the MS, particularly around clarifications in the methods/results. Readdressing the title is vital as it reads as misleading and seems as though an intervention was completed.
Detailed comments:
Title: Please consider re-wording as this reads as an intervention/promotion of improvement.
Abstract: the use of ‘voluminous’ reads oddly (line 37) – consider re-wording
Introduction:
Line 72-75 – needs a reference and need to be clear that this is in a Western society(?).
Line 100-101 – it is unclear why a rural community setting is being used? More information about this is required (some given in the conclusion but it needs to be introduced here and a rationale as to why this is important to look at needs to be included – are young people in rural communities anticipated to be particularly susceptible etc.?).
General point: The sentence structures require some more work as they make the piece slightly confusing in parts of the introduction but a good description of BDD is provided. Throughout the MS you make mention of ‘men’ and ‘women’ in relation to children. This is not terminology typically used for children. I would change to male and female throughout
Method:
Participants: Why does your age range go up to 21? This is not technically adolescence? Please provide a reason for this or omit this age from your analysis.
Line 152 – it is unclear what you mean by no longer attended paediatric medicine? More information needs to be given on why you decided that those under 15 are deemed ‘pre-adolescent’ – a large majority of children are going through puberty above Tanner 2 by this point…
General procedural point – you provide absolutely no information on how this data was collected and when these surveys were done. Were they done alone online? With someone? Were they done immediately after having their anthropometric measures taken? Did everyone do them at the same time? Where they done in a busy classroom? It is not good enough that there is not a ‘Procedure’ section included in your method as there may be issues with your data collection.
Results:
General points: Where is your descriptives table of your participants? Age/gender/anthropometric breakdowns should all be available in your paper.
Did you complete any normality testing/did you need to adjust for data at all prior to conducting your analyses? Did you bootstrap accordingly to reduce potential for Type 1 errors?
It is difficult to understand Table 1 – you do not note what the significance relates to at all? For instance, self-induced vomiting included 65 young people out of a total of 2,500 and was highly significant…significant in what way?
Your ‘dissatisfaction’ score is assumed rather than collected as a self-report measure. It is important to understand the limitation of this. Yes, it is very likely that the young person is looking at an ideal and not perceiving their own body as this ideal but you are assuming their dissatisfaction, rather than asking if they are indeed dissatisfied with their body as it presently is.
Discussion:
Line 254: you need to make it clear why it is important to look at a rural area. Is there an anticipated difference? Is there more risk? Etc.
Line 265: do you mean underestimate oneself rather than overestimate here?
Line 268: reference for this statement – also, how do you then justify having such a broad age range (up to 21 years old)? Or is this relating to this study? If so this should be in past tense and is confusing.
Line 273: the idea that males picked the larger body types because they like stronger bodies is arguably a stretch – the images do not ‘look’ muscled/stronger. It would be helpful to understand if that is how the participants perceived them.
Line 279: you make a point that the adolescent group was less dissatisfied than the ‘pre-adolescent’ group but do not provide possible reasons as to why.
Line 320: It is difficult to get a sense of limitations without knowing your methodology but it is important to note the age cut offs and how affected your results will be as a consequence of this as well as the assumed nature of your dissatisfaction score.
Reviewer 2 Report
Many thanks for giving me the opportunity to review the manuscript entitled "Self-perception and self-acceptance as tools for healthy eating behaviors promotion in adolescence: a cross-sectional study".
The study described a high prevalence of unhealthy weight-control behaviors (UWCB) among adolescents. It also demonstrated that adolescents tend to underestimate their body size and reports a dissatisfaction with the own body. Furthermore, it described an association between UWCB with the desire of achieving thinner images and with distorted body perceptions regardless of type of distortion.
Analysis of the problem was based on a sample of 2500 participants who willingly agreed to participate in the study. Statistical analysis was based on descriptive analysis and multivariate random-intercept mixed effects logistic regression. The results were presented not only in tables but also in informative graphs. The conclusions were based on the results of the study.
Reading the submitted manuscript, several questions arose and inaccuracies were noticed, which I recommend to fix.
- The title of the article does not accurately reflect the content of the article as it does not contain data on "health promotion". The title should also indicate the country.
- Keywords: "health promotion" is too common; "eating behaviors", according with limitations, was not analysed.
- 1. Study design and participants: Can the study sample be considered random? On what basis was it decided to study 2,500 participants? How many schools were selected for the sample? What percentage of all students in these schools participated in the study? What can be said about non-participants, for example, they did not want to participate in the study because they were overweight?
- 2.1. Perceptual and cognitive-affective dimensions: It is advisable to define more precisely what is "perceptual dimension" and what is "cognitive-affective dimension".
- 3. Statistical analysis: What software was used for analysis?
- Table 2. What is 1st degree/2nd degree of body dissatisfaction/distortion?
- Figures 2-4: What is depicted on the ordinate axis? If P are probabilities, then they should be in the range of 0 to 100.
Thank you for considering my opinion. I encourage authors to keep on working to improve the manuscript.
